



# Automated atmospheric profiling with the Robotic Lift (RoLi) at the Amazon Tall Tower Observatory

Sebastian Brill[1], Björn Nillius[1], Jan-David Förster[1,a], Paulo Artaxo[2], Florian Ditas[1,b], Dennis Geis[1], Christian Gurk[1], Thomas Kenntner[1], Thomas Klimach[1], Mark Lamneck[1], Rafael Valiati[2], Bettina Weber[3], Stefan Wolff[1,c], Ulrich Pöschl[1], and Christopher Pöhlker[1]

[1]Multiphase Chemistry Department, Max Planck Institute for Chemistry, Mainz, 55128, Germany
[2]Institute of Physics, University of São Paulo, São Paulo, 05508-090, Brazil
[3]Institute of Plant Sciences, University of Graz, Graz, 8010, Austria
[a]now at: Atmospheric Microphysics Department, Leibniz Institute for Tropospheric Research, Leipzig, 04318, Germany
[b]now at: Hessian Agency for Nature Conservation, Environment and Geology, 65203 Wiesbaden, Germany
[c]now at: German Weather Service, 63067 Offenbach am Main, Germany

**Correspondence:** Christopher Pöhlker (c.pohlker@mpic.de)

**Abstract.** The Amazon rain forest plays an important role in the biogeochemistry, water cycle, and climate of the South American continent and the Earth system. The Amazon Tall Tower Observatory (ATTO) has been established to study and quantify forest-atmosphere interactions under natural conditions, as well as the transformation of the Amazon ecosystem as a result of increasing perturbations related to deforestation and climate change. Here, we present the design and first results

5 of a custom-made Robotic Lift system, RoLi, installed to automatically measure high-resolution vertical profiles along the 325-meter tall ATTO tower at high spatial and temporal resolution at vertical profiling speeds up to $0.5\,\mathrm{m\,s^{-1}}$. The RoLi payload of up to $80\,\mathrm{kg}$ can be flexibly adjusted and comprises meteorological, trace gas, and aerosol instruments with short inlet lines, minimizing potential wall losses and related artifacts that may occur in longer sampling tubes of the tall tower. First measurement results show pronounced spatiotemporal patterns in the altitude profiles of temperature, humidity, fog, and

10 aerosol particle concentration and size, providing new insights into the diel interplay of convectively mixed daytime and stable stratified nighttime conditions. The RoLi data will help to better constrain the gradients and exchange of air masses, gases, and particles across the forest-atmosphere interface and related mixing processes in the lowermost planetary boundary layer.



# 1 Introduction

The Amazon basin is home to the largest rain forest in the world, which covers about $6 \times 10^6 \, \mathrm{km}^2$. It is characterized by a high biodiversity and biomass turnover, a complex hydrological cycle, and plays an essential role in the Earth system (Kueppers et al., 2004; Engle et al., 2008; Zeng et al., 2008; Pöschl et al., 2010; Melack and Hess, 2011; Pöhlker et al., 2012; Agudelo et al., 2019; Ruiz-Vásquez et al., 2020; Machado et al., 2024). Deforestation, forest fragmentation, and the effects of climate change pose significant threats to the Amazon ecosystem, as they cause habitat loss, disrupt the region's ecological balance, and weaken the forest's stability (Laurance et al., 2000; Davidson et al., 2012; Lapola et al., 2014; Tollefson, 2015; Khanna et al., 2017; Mitchard, 2018). Researchers have been trying to understand and quantify how climate change, climate extremes, and land use change influence the interactions between intact Amazonian forests and the atmosphere, along with its consequences for water cycling, biodiversity, and climate on continental and global scales (Davidson et al., 2012; Andreae et al., 2015; Wendisch et al., 2016).

The Amazon Tall Tower Observatory (ATTO) was established in 2012 about 150 km north-east of Manaus as a long-term and continuous measurement station (Davidson et al., 2012; Andreae et al., 2015; Pöhlker et al., 2019). The core of the project is the 325 m tall tower, which provides continuous measurements of atmospheric parameters at the forest-atmosphere interface and throughout the lower planetary boundary layer (PBL). Tall towers exist at several locations worldwide (Heintzenberg et al., 2011; Kohler et al., 2018; Meng et al., 2020; Dvorská et al., 2015) and provide continuous and highly time-resolved measurements at multiple heights across the lower PBL, capturing both local processes near the surface and regional signals in the lower layers of the troposphere. Tall tower measurements help to constrain the interplay of different trace gas and aerosol sources, related to (tropical) meteorology including the formation and dissipation of stable nocturnal and convective daytime boundary layers, intermittent turbulent structures, and the entrainment of air masses from higher altitudes (Fisch et al., 2004; Li et al., 2010; Oliveira et al., 2020). Complementary to smaller flux towers, which measure at or above canopy height and have a limited footprint area, tall towers have significantly larger footprints and therefore integrate regional processes (Pöhlker et al., 2019). A focal point at ATTO has been the research on atmospheric aerosols and their effects on radiative transfer (Liu et al., 2020; Morais et al., 2022; Holanda et al., 2023), cloud and precipitation formation (Pöhlker et al., 2018, 2023; Efraim et al., 2024), and ecosystem processes (Löbs et al., 2020; Prass et al., 2021b; Barbosa et al., 2022; Mota de Oliveira et al., 2022). Especially for trace gas, aerosol, and cloud cycling, tall tower measurements provide unique data to document the baseline conditions in currently still untouched regions of the Amazon, as land use and climate change continue to reshape the rain forest ecosystem.

To capture the vertical exchange and processing of gases and aerosols, measurements are typically conducted at multiple inlet heights. At ATTO, greenhouse gases are currently measured at six heights (4, 42, 81, 150, 273, 321 m), volatile organic compounds (VOCs) at four heights (i.e., 40, 80, 150, and 321 m; (Zannoni et al., 2020; Pfannerstill et al., 2021; Ringsdorf et al., 2024), reactive species such as ozone at eleven heights (i.e., 0.05, 0.5, 4, 12, 24, 36, 53, 79, 80, 150 and 320 m), and aerosols at two heights (i.e., 60, and 325 m, (Andreae et al., 2015; Franco et al., 2024; Machado et al., 2024)). However, ob-



servations at few defined inlet heights are subject to experimental limitations as they provide only a simplified representation of the four-dimensional atmospheric exchange and processing of gases and aerosols. Especially aerosol observations are often only conducted at one and sometimes at few measurement heights, due to the demanding and relatively inflexible construction of stainless steel inlet lines (Birmili et al., 2007; Center for Aerosol In-Situ Measurement - European Center for Aerosol Cali-

bration and Characterization (ECAC-CAIS), 2024). Those tubes typically have comparatively large diameters (i.e., at ATTO 1 and 1.5 inch) to ensure high sample air flow rates to supply multiple instruments and at the same time laminar flow profiles and reduced particle losses (Kumar et al., 2008; von der Weiden et al., 2009). In addition, long inlet lines are always associated with unavoidable aerosol particle losses due to diffusion, inertial impaction, sedimentation, thermophoresis, and electrostatic effects (von der Weiden et al., 2009; Heintzenberg et al., 2011). Likewise, reactive trace gases, such as sesquiterpenes, diterpenes or

amines, can hardly be sampled through long inlet lines due to wall losses and their short atmospheric lifetime in the presence of reactive species (Li et al., 2023; Deming et al., 2019).Vertical profile measurements using drones eliminate the need for long inlet lines but are largely constrained by payload capacity and flight time. In contrast, utilizing a mobile platform on tall towers for vertical profile measurements also removes the need for long inlet lines while offering significantly fewer limitations in payload capacity compared to drones (Brown et al., 2013).

In this study, we present the development of the Robotic Lift system, RoLi, along with exemplary data from its deployment at the $325\,\mathrm{m}$ tall tower at ATTO. RoLi allows to:

1. obtain vertical profiles of meteorological, trace gas, and aerosol parameters at very high spatial resolution (between centimeters and meters, depending on instrument response and speed of RoLi) between $8.3\,\mathrm{m}$ and $318.3\,\mathrm{m}$ height;

2. use the same set of instruments for measurements of high resolution profiles, which eliminates uncertainties associated

with instrument comparisons and cross-calibrations when multiple sensors are used in parallel at different heights;

3. operate instruments with very short inlets across multiple heights to minimize effects of unavoidable trace gas or aerosol particle losses in long inlet lines (e.g., sampling of ultrafine aerosol particles or highly reactive VOCs (von der Weiden et al., 2009; Li et al., 2023));

4. conduct observations and sampling at freely chosen heights to capture events of particular interest, such the nocturnal

boundary and residual layers or defined fog layers.

RoLi complements the existing experimental setups at ATTO as it serves as a flexible platform for meteorological, trace gas, and aerosol sensors. It offers highly detailed vertical profiles and enables flexible sampling of aerosols and reactive trace gases, making it a valuable addition to atmospheric research capabilities.



## 2   Technical description

### 2.1   Study site

The Amazon Tall Tower Observatory (ATTO; 2.1459° S, 59.0056° W, 134 m a.s.l.) is located in the central Amazon Basin, about 150 km northeast of the city of Manaus in the Uatumã Sustainable Development Reserve (USDR) (Andreae et al., 2015; Pöhlker et al., 2019). The ATTO project started in the year 2012, as a cooperation between Brazil and Germany. The location was chosen because of its flat terrain, the relatively pristine atmosphere during the wet season, and the good accessibility from Manaus (Martin et al., 2010; Pöhlker et al., 2016; Saturno et al., 2018; Pöhlker et al., 2023). By bridging multiple disciplines, the ATTO project aims for a better understanding of the role of the Amazon in the Earth system. The ATTO site includes three measurement towers (1 x 325 m, 2 x 80 m height) and is equipped with various instruments for meteorological, aerosol, trace-gas, and ecological research (Andreae et al., 2015; Löbs et al., 2020; Prass et al., 2021a; Corrêa et al., 2021).

### 2.2   Overall RoLi design and installation at the tall tower

The entire RoLi system consists of the main robotic lift, the tower and ground installation, and the control software. The robotic lift, which houses all instruments and sensors, moves vertically along the tower. The tower and ground installations comprise the main aluminum rail, two power rails, an energy supply unit, as well as WiFi and radio antennas. The system is operated and automated via custom-developed software that manages lift control, data storage, and safety mechanisms. This software runs on a control computer located in a ground-based container near the tower. All main components are shown in Figure 1 and are also described in detail below.

### 2.3   Aluminum rail and power rails

The backbone of the RoLi system is the 320 m long aluminum rail (HighStep Systems AG, Silbernstrasse 10, 8953 Dietikon, Switzerland), installed along the southern outside corner of the 325 m tall tower (aligned at 202° south), reaching from 0.05 m up to 320 m (Figure 1; Figure 2 A and B). The rail consists of 6 m sections, each secured to the tower structure every 3 m. This commercial rail system is ideal for RoLi, as it has been easy to install and maintenance-free. Its marine grade anodized surface provides additional protection against harsh environmental conditions. However, due to the humid conditions below the canopy, the lower 40 m of the rail require annual cleaning to remove dirt and algal growth.

The lift receives electrical power via two power rails (Stromschleifleitung 0812, Conductix-Wampfler GmbH, Rheinstrasse 27 + 33, 79576 Weil am Rhein, Germany), positioned on either side of the aluminum rail. These 4 m sections are joined to match the length of the HighStep rail, with mounting points every 4 m along the tower structure and a spacing of 128 mm from the aluminum rail. The rails carry a voltage of 230 V, derived from the voltage difference between two 115 V line conductors (L1 and L2) in a split-phase system, which is supplied at the base near the ground. As one of several safety measures, the power rails are monitored by an insulation control unit (Bender IR425-D4W-2, Bender GmbH & Co. KG, Londorfer Straße 65, 35305





Grünberg, Deutschland) that continuously measures insulation resistance against ground to ensure ground fault protection. The
system is designed to temporarily shut off when the insulation resistance drops below $100\,\mathrm{k\Omega}$ and automatically restart when
the insulation rises above this threshold. If the insulation resistance falls below $10\,\mathrm{k\Omega}$, the system shuts down permanently and
requires a manual reset. As an additional safety feature to prevent data loss during power outages or irregularities caused by
generator switches, the entire RoLi system, along with all ground installations, is connected to a UPS capable of providing up
to 20 minutes of backup power under normal load. When RoLi is removed from the rail, the power rails are grounded to serve
as lightning protection of the electrical infrastructure.

## 2.4 Robotic Lift (RoLi) hardware

The lift consists of an aluminum frame built from 40 x 40 mm aluminum profiles (MayTec Aluminium Systemtechnik GmbH,
Gewerbering 16, 82140 Olching, Germany) and custom-made aluminum components that ensure structural stability, on which
all parts are mounted (Figure 1). Its central components include three motors with gearboxes (Relex 40-01B-H04-VD-5_00,
Relex AG, Schachenstrasse 80, CH-8645 Jona), motor controllers (Platinum Bee, Elmo Motion Control GmbH, Walter-
Oehmichen-Str.20, 68519 Viernheim, Germany), and metal gears, which connect to the aluminum rail and drive RoLi up and
down. These engine-gearbox units contain 48 VDC motors, each with a rated power of 630 W, a nominal speed of 6000 rpm,
and a peak torque of 4.41 Nm. The motors are connected to gearboxes with a total gear ratio of 1:40 (with a planetary gear
stage ratio of 1:8 and a worm gear stage ratio of 1:5), yielding a maximum torque of 44.5 Nm. Each motor unit also includes
a 24 VDC electromagnetic coupling brake with a holding torque of 7.5 Nm, which deactivates while in motion and engages
when the lift stops, securely holding RoLi in place without using motor force.

The power of the engine-gearbox units is transmitted to the aluminum rail via aluminum bronze alloy gear wheels. Replaceable
plastic caps (HighStep Systems AG, Silbernstrasse 10, 8953 Dietikon, Switzerland) on the gear wheels are used to extend the
lifespan of both the rail and the gears. The durability of these plastic caps depends on payload and environmental conditions
(such as rail dryness and cleanliness). With a typical payload of around 70 kg and 48 profiles per day, the caps require weekly
replacement, corresponding to 336 full profiles at the ATTO tower and 107 km of vertical travel. Replacements have been made
proactively to prevent operational failure. While descending, an actively fan-cooled magnetic powder brake (Series B.651.V,
IBD Wickeltechnik GmbH, Böllingshöfen 79, 32549 Bad Oeynhausen, Germany) dissipates the potential energy from RoLi as
heat. Relying solely on the motors for descent could damage the power supply and other 48V electronics. The brake allows for
continuously adjustable braking force of up to 65 Nm of torque and is nearly maintenance-free.

Two current collectors (Stromabnehmer 0812 1P2PE 98A16 REV B, Conductix-Wampfler GmbH, Rheinstr. 27 + 33, 79576
Weil am Rhein, Germany) with two connection points each provide uninterrupted 230 VAC power supply, also across power rail
connection points or during brief interruptions caused by dirt on the rail. Power from the rails directly supplies the scientific
payload and is converted by an onboard power supply (PHP-3500-48, MEAN WELL USA, INC., 44030 Fremont Blvd.,





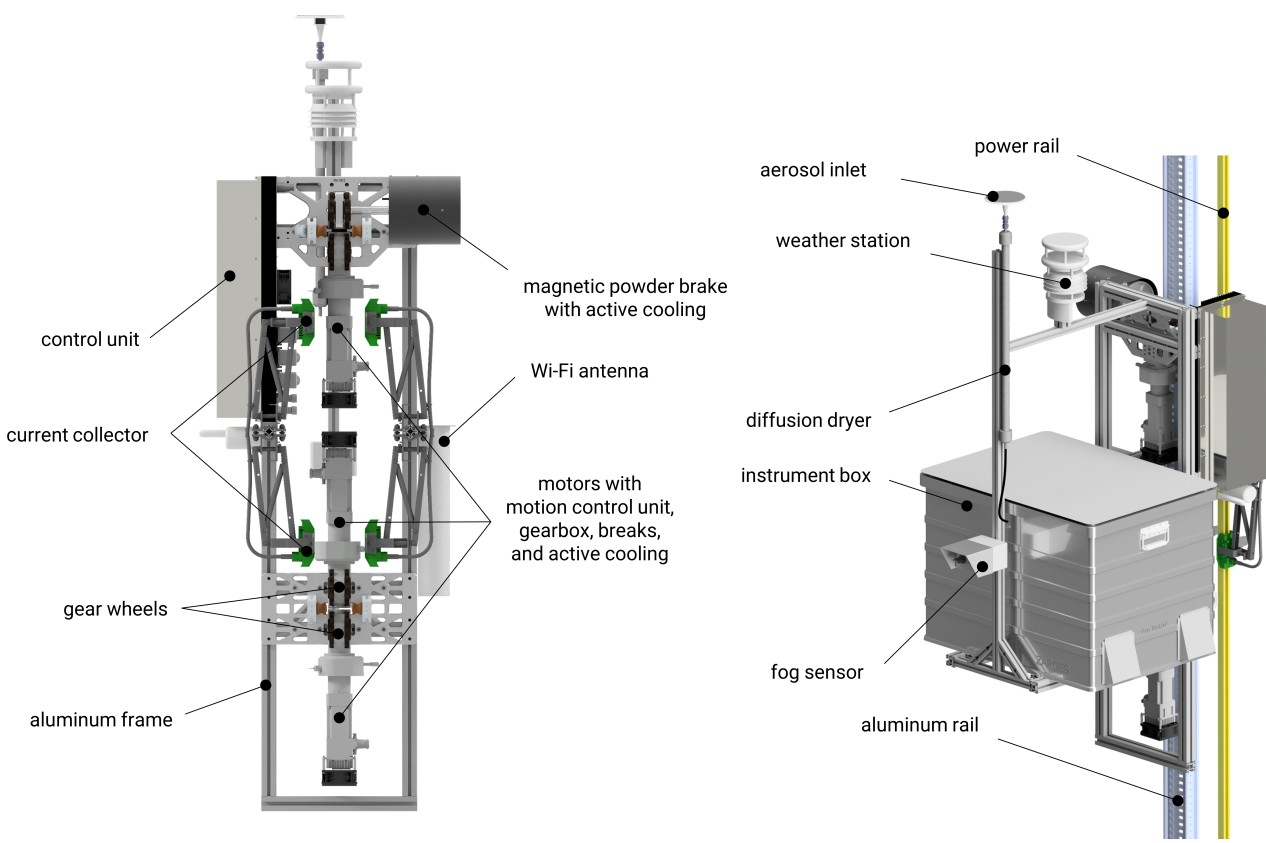

**Figure 1.** Overview of the major RoLi components. The rendered view on the left shows the rear of RoLi, with the major components labeled, while the right view illustrates the front, including the mounted instrument box and tower rails. During operation, the four current collectors, with two on each side, are connected to the power rails, which operate at 230 VAC. This voltage is directly utilized to power the scientific instruments. Within the control unit, an internal power supply transforms the voltage to 48 VDC to supply the RoLi subsystems, particularly the motors. The three motors, along with their gearboxes and brakes, are securely mounted onto the RoLi frame. The instrument box is fully modular and can be easily removed if needed. The aerosol inlet head is positioned on top of the diffusion dryers, while the pumps used for creating the vacuum needed for sample air dehumidification are housed inside the instrument box. The weather station is also mounted on the aluminum frame, directly above the instrument box, ensuring seamless integration with the lift system. The main connection to the control unit on the ground is via WiFi.

135  Fremont, CA 94538, USA) to 48 VDC to drive the motors. To increase safety the power rails are connected to the mains through an isolation transformer (TG1-6,0; Walcher GmbH, 36124 Eichenzell).



To ensure precise data collection and safety, it is crucial to keep track of the exact position of the lift and to detect the rail limits. Several anchor points are set on the upper and lower ends of the aluminum rail. Detection of these points is achieved with four inductive proximity sensors (Induktiver Sensor NBB10-30GM60-A2-V1, Pepperl+Fuchs Vertrieb Deutschland GmbH, Lilienthalstraße 200, 68307 Mannheim, Germany), two installed on each side of RoLi. These sensors continuously monitor the distance to the aluminum rail, triggering when distances change. At each reference point, metal markers alter the sensor distance, triggering the internal software. These proximity sensors detect specific rail points, and once positioned, the lift's movements during profile measurements is logged based on gear wheel rotations. The rail between the ground and 320 m has three main reference points. The lowest is the "Dock" point, located at approximately 4 m, which aligns with the height of the roof above the ground-based measurement containers next to the tower. This position is used for routine safety inspections and instrument checks. For profile measurements, the lowest operational point is set at the "Home" point, located at 8.3 m, for safety reasons. Using the proximity sensors on the left side, RoLi detects whether it is above or below the Home point. The third reference point, at 318.3 m, marks the uppermost position on the rail, indicating the maximum height achievable. These reference points are sensed by the left-side proximity sensors, while the right-side sensors act as additional hard-coded limits to enhance safety. The signals from the proximity sensors are processed by a Teensy (PJRC.COM, LLC., 14723 SW Brooke CT, Sherwood, OR 97140, USA), which then sends the commands directly to the motor controllers. This was implemented as an additional safety feature, so that even if the control software commands RoLi to move beyond 318.3 m or below 4 m, the electronics would prevent such movements. These limits cannot be overridden by the control software.

While commands and data storage are managed by the control computer in the ground-based container, the driving software that issues commands and monitors safety operates on an onboard Raspberry Pi IoT Gateway (Axotec Technologies GmbH, Sudetenstraße 88, 82538 Geretsried, Germany). Commands from the ground station are processed by the onboard Raspberry Pi computer, which then controls the motors and brakes. Signals from the proximity sensors trigger safety protocols in the Teensy or set anchor points directly to the onboard computer. In essence, the control computer in the ground-based container defines movement targets, while the onboard Raspberry Pi computer and the Teensy translates these into motor actions.

## 2.5 Remote access and data transfer

To control the lift, transfer data, and remotely access the instruments, a stable connection to the lift is essential. Two independent systems are used to ensure a consistently stable connection and safe operation. The primary system for connecting the ground installations with the lift involves a Wi-Fi connection (2.4 GHz), utilizing a directional Wi-Fi antenna (LogiLink® Wireless LAN Antenne Grid Parabolic 24 dBi, 2direct GmbH, Langenstück 5, 58579 Schalksmühle, Germany). The parabolic antenna is installed on top of the protective roof of the ground-based containers near the ATTO Tower, pointed upwards to the top of the tower, covering the entire range between the ground and the tower's top. The Wi-Fi network dedicated to RoLi is not shared with other devices on the tower, ensuring uninterrupted communication. The Wi-Fi antenna is connected to an access point (airMAX Rocket M2, Ubiquiti Inc., 685 Third Ave, Ste 27, New York, NY 10017, USA), which in turn connects to the control computer running the RoLi software. RoLi itself is equipped with a long-range Wi-Fi antenna (Cyberbajt YAGI 24-16





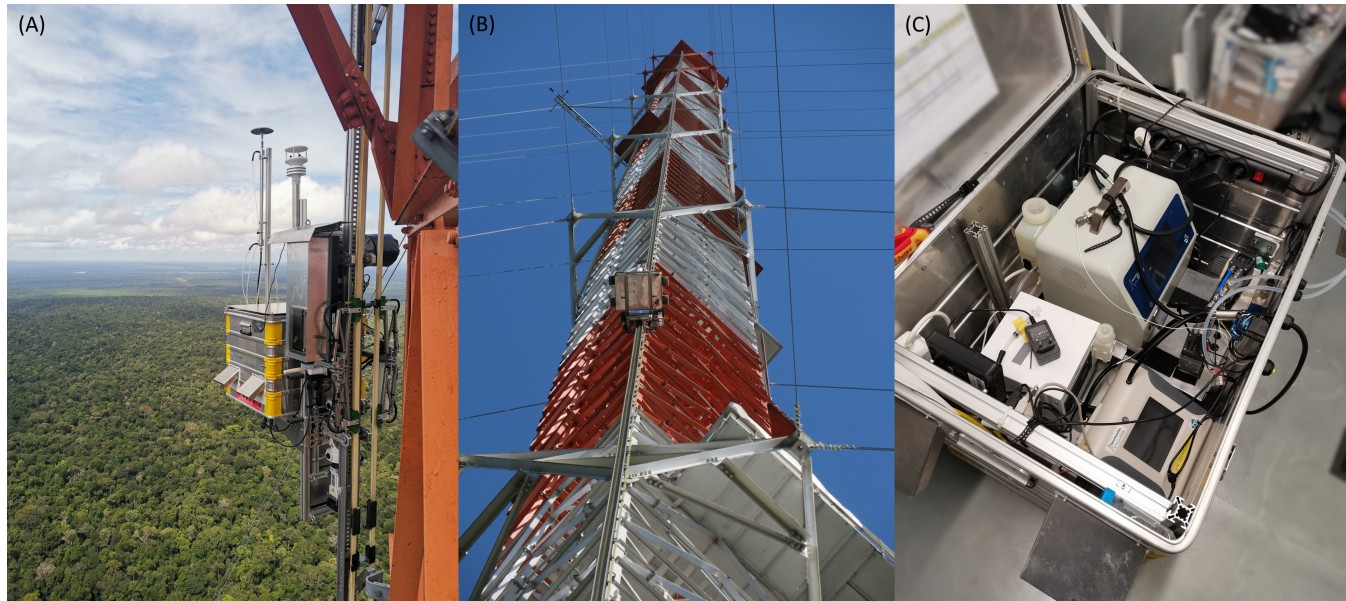

**Figure 2.** Pictures of RoLi with its instrument box during operation at the tall tower at the ATTO site. (A) RoLi in operation at the uppermost position at the tower, 318.3 m above ground. (B) RoLi climbing the tower. (C) Instrument box containing the TSI NanoScan SMPS (upper left), Aerosol Devices MAGIC CPC (lower left), TSI Optical Particle Sizer (lower right), DMT WIBS-4A Bioaerosol Sensor (not visible, below CPC and OPS), 2B Tech Ozone Monitor (not visible, below SMPS), measurement computer, and pumps for the diffusion dryer.

- 2.4 GHz Wi-Fi Directional Yagi Antenna, Cyberbajt, Młyńska 27, 22-400 Zamość, Poland) to maintain its connection to the network (Figure 1). On RoLi, the Wi-Fi antenna is connected to a WLAN client (Moxa AWK-1137C, 13F, No. 3, Sec. 4, New Taipei Blvd., Xinzhuang Dist., New Taipei City 242, Taiwan), which links the instrument box and the RoLi internal computer to the Wi-Fi network. As a safety measure to ensure a continuous connection to RoLi, a fallback radio connection operates in parallel with the main Wi-Fi connection. At the base, a 14 dBi 433 MHz radio antenna (WY 400-10N, SIRIO Antenne S.r.l., Via

Liguria 15, 46049 Volta Mantovana, Italy) connects to RoLi (Delock 433 MHz Antenne N Stecker 1.45 dBi, Tragant Handels- und Beteiligungs GmbH, Beeskowdamm 13/15, 14167 Berlin, Germany) via a radio signal. This backup connection activates only if the Wi-Fi connection fails, ensuring that the lift can still be remotely operated at all times.

## 2.6 Software and automation

The control and data acquisition functions are managed by the "VBUS system", a specialized platform developed at the Max

Planck Institute for Chemistry (MPIC). This software operates on a computer housed inside a container at the base of the tower. The computer is permanently connected to the RoLi network, facilitating command transmission to the lift and receiving scientific data. Additionally, it is integrated into the ATTO local network for time synchronization and remote access. The VBUS system consists of electronic modules based on microcontrollers and offers a flexible software environment, including





scripts and a graphical user interface (GUI). A representative screenshot of the GUI is provided in Supplement Figure A1. The
lift movement is entirely controlled by this software. All operations can be executed using four primary commands:

– **Homing**: When the lift is without power, it loses its positional reference on the tower. If the position is unknown, the only
accessible command is "Homing". This command directs the lift to move upward (if it is below the "Home" point) or
downward (if it is above the "Home" point). The lift determines its relative position based on the detection of proximity
markings on the aluminum rail. During the "Homing" process, the maximum speed is limited to $50\,\mathrm{mm\,s^{-1}}$ if RoLi is
above the "Home" point and to $10\,\mathrm{mm\,s^{-1}}$ if RoLi is below the "Home" point. Upon reaching the Home point, the height
is automatically set to zero meters, establishing the reference point for measurement mode. Once set, the Home point
does not need to be recalibrated unless the lift is powered off. Position tracking during operation is continuously updated
via the rotation of the gear wheels.

– **Go to Position**: During normal operation, every change in position is executed with the software command "Go to
Position". The command requires specifying the target position (from $0\,\mathrm{m}$ to $310\,\mathrm{m}$ in software terms, corresponding
to $8.3\,\mathrm{m}$ to $318.3\,\mathrm{m}$ actual height) and the speed at which the lift should move. While the lift's maximum speed is
$500\,\mathrm{mm\,s^{-1}}$, the typical speed used is $333\,\mathrm{mm\,s^{-1}}$ to minimize material wear and reduce vibration that could affect the
instruments.

– **Go to Dock**: For routine maintenance and instrument checks, the lift must be accessible to users. The dock point is set at
$4\,\mathrm{m}$ above ground, approximately the height of the container roofs, allowing users to access the lift by stepping onto the
container roof. For uninstalling RoLi, it must be manually moved past the software limit of $4\,\mathrm{m}$. This requires manually
deactivating RoLi's brakes, allowing it to "slide" slowly down the rail in a slow and controlled way. Once below the $4\,\mathrm{m}$
limit, the lift can again be controlled with the RoLi software but at a reduced maximum speed of $10\,\mathrm{mm\,s^{-1}}$.

– **Stop**: In case the lift needs to be stopped, for example, in case of an emergency, the "Stop" command can be used to
immediately halt movement and lock RoLi in its current position. Using the "Stop" command also disables automation.

To streamline operations and reduce manual input, an automation feature was added to the software. This feature allows users
to program a sequence of commands, enabling automated measurements for extended periods. The number of commands and
the duration of the automation mode are unrestricted. However, for safety reasons, it is recommended to pause the automation
program at least once daily to perform a hardware inspection of the lift. During automation, the same height and speed limits
apply as in manual mode ($0\,\mathrm{m}$ to $310\,\mathrm{m}$ in software terms, corresponding to $8.3\,\mathrm{m}$ to $318.3\,\mathrm{m}$ actual height). The software
also continuously receives and logs data from RoLi during operation. This includes housekeeping data such as position, speed,
motor temperatures, brake temperature, and various electronic parameters. Additionally, data from the weather station such as
temperature, humidity, pressure, wind speed, and wind direction are also logged. Instrument data are recorded separately by
the onboard instrument computer. To ensure synchronization between the lift and instrument data, accurate time alignment is
essential. The control computer, and the computers on board of RoLi are synchronized with the ATTO time server once per
hour to minimize time shift. The software also facilitates remote access to the instruments installed on RoLi, allowing users



to monitor instrument parameters or check RoLi's operational status in real-time. This functionality enhances efficiency and oversight during operation.

To further enhance safety during operation, especially when the lift is in automatic and unattended mode, several safety features
have been implemented in the control software on RoLi:

– **Wind Alarm**: The wind data from the onboard weather station are used to trigger a wind alarm if the wind speed exceeds $15\,\mathrm{m\,s^{-1}}$. If this value is exceeded for more than 5 seconds, a wind warning is automatically triggered in the software. This immediately stops automatic mode, disables user inputs, and sends the command "go to home" which is executed at a speed of $-333\,\mathrm{mm\,s^{-1}}$.

– **Connection Alarm**: If the Wi-Fi connection to the grounds station is lost, a connection error is triggered and RoLi automatically drives to the home position at $333\,\mathrm{mm\,s^{-1}}$, and all user inputs, as well as automatic mode, are deactivated.

– **Temperature Alarm**: If the temperature of one of the motors or motor controllers exceeds $70\,°\mathrm{C}$, a temperature warning is triggered. Above $60\,°\mathrm{C}$, the cooling fans mounted on top of the motor control unit are activated to prevent further temperature increases. During normal operation, the temperatures of the motors and motor controllers always remain
below $70\,°\mathrm{C}$. If the temperature warning is triggered, the lift immediately stops all operations and locks in place, waiting for the motors to cool down.

Additionally, in case of an obstacle blocking the track of RoLi, for example dirt on the aluminium rail, the on-board electronics will detect the increased unusual torque of the motors and stop the movement and disable user inputs.

Whenever any warning is triggered, all user inputs are blocked, and automation is turned off. To resume operation, a user must
manually clear the warning message in the RoLi control software after checking the cause of the warning and ensuring safe operation. Overwriting warning messages is possible under specific circumstances using the "Homing" command to ensure RoLi can always return to its base position.

## 2.7 Instrument box and payload

The current version of RoLi is designed to house all scientific instruments inside a modified aluminum box (K 470 Universalk-
iste, ZARGES GmbH, Zargesstraße 7, 82362 Weilheim, Germany) measuring $800\,\mathrm{mm} \times 600\,\mathrm{mm} \times 610\,\mathrm{mm}$ (Figure 1 and 2). The box has been modified with two angle plates at the bottom to create a secure connection with the lift. To prevent vibrations of the lift from being transferred to the instrument box, the entire box is mounted on rubber vibration absorbers. To prevent overheating inside the box, four ventilation openings have been added to its sides. Two 24 V fans are installed on one side to provide strong ventilation. For additional insulation from intense solar radiation, a metal sheet has been installed on top of the
box, with an air gap between the box and the sheet. This setup prevents the top of the box from heating up in direct sunlight. At the base of the metal box, two waterproof connectors have been installed to provide power and network connections inside the box. The power supply into the box is 230 VAC, suitable for most scientific instruments. A network connection is necessary to



remotely access the instruments, allowing for data quality maintenance during lift operation and time synchronization with the local time server. To ensure safe operation in wet conditions, the metal box is grounded to the metal frame of RoLi, which, in

turn, is grounded through its connection to the aluminum rail.

The instrumentation used for data collection is modular and can be compiled according to the research question(s). Limitations for the scientific instrumental setup include the weight that can be carried and the available space inside the metal box. The maximum weight that can safely be carried has been tested to be 80 kg. The limiting factor is the braking system, which holds RoLi in place when the motors are turned off. Each brake can hold approximately 85 kg, which would theoretically allow for

a maximum weight of 255 kg (Robotic Lift (RoLi) hardware 130 kg, theoretical maximal payload 125 kg). With an additional safety margin, the maximum payload has been set to 80 kg. In addition to the scientific instruments, the instrument box also houses the vacuum pumps for the diffusion dryer, a flow controller to regulate the dryer, a temperature sensor to monitor the internal temperature of the box, and a computer running the software necessary to operate the instruments. The aerosol flow is dried using a diffusion dryer (MD-700-24S-3, Perma Pure LLC, 1001 New Hampshire Ave., Lakewood, NJ 08701, USA). The

humidity of the sample flow is continuously monitored and logged by a humidity sensor (MSR145, MSR Electronics GmbH, Mettlenstr. 68472, Seuzach, Switzerland) installed in the sheath air flow of the SMPS. To prevent rainwater and insects from entering the aerosol flow, a custom-made aerosol inlet head with a metal mesh has been installed on top of the diffusion dryer.

During the first campaigns of RoLi, the following set of instruments has been used:

– Optical Particle Sizer (0.3 – 10 µm; TSI Incorporated, 500 Cardigan Road, Shoreview, MN 55126, USA)

– NanoScan SMPS (10 – 420 nm; TSI Incorporated, 500 Cardigan Road, Shoreview, MN 55126, USA)

– MAGIC CPC (5 nm – 2.5 µm; Aerosol Devices, 1613 Prospect Park Way, Fort Collins, CO 80525, USA)

– WIBS 4A Bioaerosol Sensor (>1 µm; Droplet Measurement Technologies, 2400 Trade Centre Avenue, Longmont, CO 80503, USA; not shown in this study)

– Ozone Monitor (2B Technologies, 6800 W. 117th Avenue, Broomfield, Colorado 80020, USA)

– Aethalometer AE33 (Magee Scientific, 1916A Martin Luther King Jr Way, Berkeley, CA 94704, USA; not shown in this study)

In addition to the instruments installed inside the box, two instruments have been mounted externally on the RoLi frame. On the front side, pointing away from the tower and in a horizontal orientation, a fog detector (Eigenbrodt GmbH & Co. KG, Baurat-Wiese-Strasse 68, 21255 Königsmoor, Germany) has been installed to measure visibility. To measure basic meteoro-

logical parameters such as temperature, humidity, pressure, wind speed, and wind direction, a weather station (METSENS500, Campbell Scientific, 80 Hathern Road, Shepshed, Loughborough LE12 9GX, United Kingdom) has been installed on the RoLi frame, above the instrument box (Figure 1). Precipitation data were obtained from a weather station (WS600-UMB, OTT HydroMet Fellbach GmbH, Gutenbergstr. 20, 70736 Fellbach, Germany) installed at an elevation of 325 m at the ATTO Tower.







**Figure 3.** Exemplary RoLi housekeeping data for 24 hours of operation on 08 Jan 2023, illustrating the detailed observation and documentation of the system performance. The regular movement pattern followed a daily cycle of 48 profiles, with two profiles measured per hour at a speed of $333\,\mathrm{mm\,s^{-1}}$. Each profile took approximately 15 minutes to be measured. Between profiles, there was always a waiting period of around 15 minutes to allow the motors and brakes to cool down. Around 09:00 local time, a daily hardware check was performed, which involved driving RoLi down to the dock position and then back up to the home position. (A) Position data of RoLi and commands sent by the software. (B) Temperature data for the three motors M1, M2, and M3 (also reflecting the temperatures of the gearbox and parking brake). (C) Temperature of the magnetic powder brake. (D) Ambient relative humidity measured with the weather station. (E) Relative humidity of the sample air measured at the SMPS sheath flow.




# 3 Performance evaluation and selected data

## 3.1 System performance

A challenge in constructing the automated lift system for high-resolution vertical profile measurements was designing it to withstand the harsh conditions of the rainforest. The high relative humidity and heavy rainfall, particularly during the wet season, posed a considerable risk of rust formation and water condensation inside the tubes and instruments. To address the issue of water condensation, a diffusion dryer was installed as a preventive measure. This ensures that the sample air

for the instruments consistently maintains a low relative humidity. Even during the night and early morning, when ambient relative humidity levels approach 100 %, the sample air relative humidity remained below 40 % (Figure 3 E), in line with established recommendation for *in situ* aerosol sampling (Center for Aerosol In-Situ Measurement - European Center for Aerosol Calibration and Characterization (ECAC-CAIS), 2024). Rain events generally did not pose a risk to the RoLi system, as all components were designed to be waterproof or protected against rain. The instrument box is inherently waterproof, and

the openings for the cooling fans were shielded by protective metal covers.

One challenge for the system was maintaining insulation of the power rails against the ground. The ground fault insulation control unit continuously monitors insulation levels, which were typically around $1.9 \times 10^7 \, \mathrm{k\Omega}$ under dry conditions. However, during the early morning, when humidity condensed on the rails, or during heavy rain events, the insulation dropped significantly due to leakage currents flowing to the tower across the long rail distance. During very heavy rain events, the system

occasionally experienced temporary power outages when the ground fault insulation dropped below $100 \, \mathrm{k\Omega}$.

During normal operation, 48 profiles per day were measured with a payload of around 70 kg (Figure 3 A). Upward profiles were always scheduled to begin on the hour and were completed about 15 minutes later at the top of the tower, driving a speed of $333 \, \mathrm{mm \, s^{-1}}$. The 15-minute waiting time at the top of the tower allowed the motors to cool down and provides an opportunity for cross-calibration of the RoLi instruments with the permanently installed instruments at the top of the ATTO

tower. After the 15-minute waiting period, downward profiles were started, followed by another 15-minute waiting time before starting the next upward profile on the following hour. Especially during daytime, high ambient temperatures and intense solar radiation posed limitations due to the risk of overheating RoLi components during operation. This issue was particularly critical during upward movements, as the motors generate significant heat that must be effectively dissipated to prevent overheating. During downward movements, motor heating was less of a concern compared to upward movements. However, the potential

energy from the high elevation must still be dissipated with the magnetic powder brake. The chosen movement pattern proofs to provide sufficient time for the motors and brakes to cool down during the waiting period as the maximum motor temperature appears to consistently remain below 60 °C (Figure 3 B and C).

For data collection, a schedule of 48 profiles per day provides a reasonable time resolution, as most atmospheric processes occur on timescales longer than this interval. The vertical resolution depends on the measurement intervals of the instruments,




which were 1 Hz for the meteorological data, 5 seconds for the OPS and CPC, and 1 minute for the SMPS and fog sensor. These intervals offer sufficient resolution to capture the relevant atmospheric variations.

### 3.2 Meteorological observations

To show the potential of the RoLi system, a period of four consecutive days during the wet season of 2024 (April 24 to April 27) were analyzed. A diurnal temperature cycle is clearly visible in the data, with the highest temperatures occurring
around the canopy level ($\sim 30$ m at ATTO (Helmer and Lefsky, 2006)) in the afternoon (Figure 4 B). Temperatures on April 26 and April 27 were significantly higher (26.2 °C) compared to April 24 and April 25 (23.9 °C), mainly due to higher cloud cover during the earlier days. Precipitation was recorded on all four consecutive days, with the heaviest rainfall occurring around noon on April 25 (34.9 mm) and lighter rain events on April 24, April 26, and April 27 (6.5 mm, 1.2 mm, and 1.4 mm, respectively).

The impact of the heavy rainfall on April 25 is evident across all meteorological parameters measured with RoLi, particularly in the wind profile data (Figure 4 D). During the event, wind speeds reached up to 11.6 m s$^{-1}$ at 318.3 m height, primarily due to downdrafts from convective clouds. This event was also marked by a steep temperature drop, further illustrating the influence of strong rain events. Given that such rain events can be short-lived, maintaining a short time interval between profiles is crucial. The schedule of 48 profiles per day appears sufficient to capture rapid changes in atmospheric properties, as demonstrated by
this example.

In the horizontal wind speed data, the nocturnal low-level jet, which occurs during most nights in the Amazon region, is clearly visible. It is characterized by strong wind speeds at night, typically increasing throughout the night and reaching a maximum around sunrise (Anselmo et al., 2020). The altitude of the maximum wind speed during low-level jet conditions correlates with the nocturnal boundary layer (NBL) top. The NBL top cannot be measured directly using the instruments installed on RoLi,
as sensible heat flux data are necessary to determine its exact height. However, an estimation is possible using the profiles of potential temperature. The effect of the strong precipitation event on April 25 is also evident in the wind direction, which shifted significantly from 250.9° in the 12 hours before the rain to 54.0° in the 12 hours following the rain (Figure 4 E). At ATTO, the predominant wind direction varies slightly, shifting from northeast during the wet season to east during the dry season. The wind direction and wind speed profiles clearly highlight the differences between daytime and nighttime conditions. During the
day, strong thermal convection generates turbulence, which is reflected in the profile data. In contrast, at night, the low-level jet produces a more laminar wind flow, resulting in smoother profiles and a more distinct wind direction.

Fog was observed on all four days shown, particularly during the nighttime hours (Figure 4 F). The most extensive fog occurred during the night of April 24 to April 25, extending from ground level to the top of the tower. In contrast, during the subsequent two nights, fog was primarily confined to the lower levels, reaching up to approximately 200 meters, with the highest occurrence
just above the canopy. A characteristic feature of fog behavior is its upward movement in the early morning, as solar heating initiates convection, leading to the formation of the convective boundary layer (CBL). The development of the CBL lifts the



fog upwards, as clearly depicted in Figure 4 F. Additionally, the strong rain event on April 25 and the minor rain event on the afternoon of April 26 are evident in the visibility data. Visibility decreases not only due to fog but also as a result of biomass burning smoke or heavy rainfall, as seen during these events.

## 3.3 Selected aerosol profiles

For the selected days of measurements, the aerosol data exhibit a highly diverse distribution pattern. During the night and early morning of April 24, a layer with a high particle number concentration was observed between 100 m and 280 m (Figure 5 C and D). Interestingly, this particle layer was poorly detected by the stationary measurements at 60 m and 325 m, as these heights were either below or above the layer (Figure 5 B). Notably, the majority of particles in this layer were in the <30 nm size range. Wind direction data indicate a pronounced wind shear in the profile during the night and morning of April 24. The layer above the canopy, up to approximately 100 m, was dominated by northwestern winds, whereas winds above 100 m predominantly originated from the northeast. The origin of these particles cannot be clearly identified, as black carbon (BC) data from 60 m were unavailable during this period. The BC data from 325 m do not indicate elevated concentrations, suggesting a predominantly natural origin. Additionally, pollution from urban areas or fires is unlikely, as the northeast wind direction does not pass over populated regions or areas affected by deforestation. Furthermore, smoke pollution would typically be evident in the aerosol mass concentration data of the OPS; however, no increased values were observed when the high-concentration particle layer was passed. With the onset of solar radiation in the morning and the initiation of convection, the layering was mixed and no longer observed.

Just before sunrise, at approximately 05:30 local time, during the downward profiling, a very thin layer of high particle concentration was detected between 50 m and 70 m, just above the canopy (Figure 5 C and D). As observed in the <30 nm size range of the SMPS data, these particles were predominantly very small. The wind direction at this time briefly shifted to the south-southwest, aligning perfectly with the direction of the on-site power generator. Despite the generator being located more than 2 km away from the tower, it cannot be ruled out that these high concentrations originated from generator emissions. Pollution from the generator is rarely detected at the tower due to the prevailing northeast wind direction, which typically prevents diesel exhaust from reaching the site. However, under specific conditions, particularly during stable NBL scenarios, an undercurrent can form within the NBL due to local meteorological conditions, potentially transporting pollution to the site where it can then be measured.

The strong rain on April 25 surprisingly did not have a strong influence on the particle population. In the morning before the rain, a layer consisting mostly of small particles passed the tower above 200 m and was subsequently mixed downward with the onset of convection. This layer was clearly visible in the CPC and BC measurements at 325 m, and with a time delay, also at 60 m (Figure 5 B). During the period of heavy precipitation, lower particle concentrations were observed, primarily due to the wet deposition of aerosol particles. However, after the heavy rain, the particle concentration and size appeared to be nearly unaffected, as surrounding air may have been rapidly entrained into the area where the precipitation had passed.



A conspicuous feature in the graph is the period between 12:00 LT and 18:00 LT on April 26, during which particle number
concentration suddenly increased across all heights, rising from $130\,\mathrm{cm^{-3}}$ to $363\,\mathrm{cm^{-3}}$, before decreasing again to $216\,\mathrm{cm^{-3}}$.
The profiles appear relatively uniform, with no strong differences between the heights, likely due to convection preventing
the formation of layers (Figure 5 C - E). A major part of the particles present during this period originated from the <30 nm
size range ($113\,\mathrm{cm^{-3}}$ between 12:00 LT and 18:00 LT). In the six hours before and after this period, the sub-30 nm particle
concentration ranged between $25\,\mathrm{cm^{-3}}$ and $37\,\mathrm{cm^{-3}}$, respectively. The median particle diameter during this event decreased
from 58 nm to 52 nm, and then increased again to 70 nm after the event. The cause of the particles responsible for this event
is unclear. Pollution from fires or urban areas is unlikely, as BC measurements from 60 m do not show any increase during
the event and were generally low (below $0.05\,\mathrm{\mu g\,m^{-3}}$) during this period. Meteorological parameters show no change in
temperature, humidity, wind direction, or wind speed, indicating no air mass change. One possible source could be the outflow
of nearby convective clouds, as a light rain passed over the tower in the evening of April 26.

During the night and morning of April 27, a strong stratification of aerosol particle number concentration occurred, with higher
concentrations observed above 100 m and comparatively lower concentrations below 100 m. Notably, at the beginning of the
event around midnight, nearly all particles were smaller than 30 nm, which is also clearly visible in both the median SMPS
particle diameter and the OPS mass concentration data (Figure 5 D - F). For these particles, pollution appears unlikely, as BC
values were consistently below $0.05\,\mathrm{\mu g\,m^{-3}}$ (Figure 5 A). Therefore, a natural origin must be considered.

Throughout all four days of the exemplary data shown in Figure 5, a clear day-night cycle is visible in the OPS total mass
concentration data. During all nights, a clear accumulation of coarse-mode aerosol particles can be observed below 50 m. This
is most likely a result of nocturnal biogenic particle emissions below the canopy, combined with the stable stratification of the
NBL above the forest, which prevents convection from mixing the particles emitted below the canopy (Figure 5 F). A majority
of these particles are likely fungal spores, as most fungi are known to emit their spores during nighttime (Gilbert and Reynolds,
2005; Elbert et al., 2007; Huffman et al., 2012; Oneto et al., 2020; Löbs et al., 2020). In the morning, when convection begins,
the accumulated particles are mixed upwards, where they can participate in fog formation and other atmospheric processes.





**Figure 4.** Vertical profile and precipitation data from April 24 to April 27, 2024: (A) Precipitation and temperature data recorded by the weather station at 325 m on the ATTO tower. (B-G) Vertical profile data measured with the robotic lift, including (B) temperature, (C) specific humidity, (D) horizontal wind speed, (E) wind direction, and (F) visibility. The impact of the rain event on April 25 is clearly evident across all parameters. Notably, the wind direction shifted from north to east during and after the rain. Additionally, fog formation is observed during the night, particularly prior to the significant rainfall event. The data gap on April 24 was caused due to RoLi maintenance.



**Figure 5.** Vertical profile, BC and particle number concentration data from April 24 to April 27, 2024: (A) BC mass concentration measured at 60 m (at the 81 m Tower) and 325 m with a Multi Angle Absorption Photometer (Model 5012 MAAP, Thermo Scientific, 27 Forge Parkway, Franklin, MA 02038) (B) Total particle number concentration measured at 60 m (at the 81 m Tower) with the MAGIC CPC (Aerosol Devices, 1613 Prospect Park Way, Suite 100, Fort Collins, CO 80525, USA) and at 325 m with the GRIMM CPC 5414 (DURAG GROUP, Headquarters, Kollaustraße 105, 22453 Hamburg, Germany) (C-F) Vertical profile data measured with the robotic lift: (C) Total particle number concentration measured with the MAGIC CPC (D) Particle number concentration below 30 nm, measured with the NanoScan SMPS (E) Median particle diameter, measured with the NanoScan SMPS, and (F) Particle mass concentration, measured with the OPS. The two fixed inlet measurement heights at 60 m and 325 m from the tower are marked as horizontal dashed lines in the vertical profile panels.



### 3.4 Tower and measurement interferences

Even though the influence of the tower structure and RoLi on the measurements should be minimal, several artifacts were observed during operation. The temperature profiles exhibit a striped pattern, suggesting some influence from RoLi on the

temperature measurements (Figure 4 A). This is primarily due to the fact that the weather station is located on top of the RoLi structure, directly above the instrument box. As a result, heat produced by the instruments and radiational heat from the sun, which warms up the instrument box during the day, can affect the temperature measurements. Additionally, the dissipating heat from the motors and the magnetic powder brake on the way down could influence the temperature measurements. On the way upwards, RoLi always encounters "fresh" air, whereas on the way down, it passes through the heat from the RoLi components

and instruments box, leading to higher temperatures being measured by the weather station. This effect is more pronounced during the daytime and under low wind conditions, as strong horizontal winds prevent the heated air from reaching the weather station. This, in turn, also affects the specific humidity data, as they are calculated from temperature values. Installing the weather station farther from the RoLi structure was considered but ultimately rejected, as this would increase the force on the gears and rail during windy conditions due to the lever effect. When analyzing the data, an alternative approach would be

to use only the upward profiles, as they are less likely to be influenced. However, for the sake of completeness, all profiles are shown in this study. Moreover, the temperature recorded by the weather station consistently lags behind the actual air temperature, especially during sudden temperature changes, as the weather station requires time to acclimate due to its specific heat capacity. When accurate temperature data are required, the use of the permanently installed weather stations across the tower is recommended.

As RoLi is installed at the southern corner of the ATTO tower, the wind data may be influenced by the tower structure when the wind is coming from certain directions. During the daytime, when solar radiation induces convection, this effect is negligible. However, at night, when thermal convection is absent, the wind flows more laminar past the tower structure, creating turbulence that can be observed in the wind speed data. This is particularly noticeable during the nights shown in Figure 4 D , when the wind is coming from the east. The anchor points of the guy wires generate horizontal artifacts in the wind speed data, which

correlate perfectly with their position on the tower (54 m, 108 m, 162 m, 216 m, 270 m, and 321 m; the last one is not visible in the data). When the wind is coming more from the north, the turbulence from the tower has a more pronounced effect on the wind measurements from RoLi, as the entire structure is passed by the wind. In the wind data, this creates additional artifacts, sometimes causing the tower structure to become visible in the wind speed data, as seen on April 25 between 18:00 LT and 23:00 LT (Figure 4 D).

The visibility measurements from the fog sensor should be unaffected by the tower structure and trees, as the measurement direction of the fog sensor points away from the tower, not in the direction of the guy wires. Additionally, the trees do not create any interference when RoLi is below the canopy, as the visibility measurement only occurs in the first few meters in front of the device and is then extrapolated. Furthermore, all aerosol and ozone data do not show any measurement artifacts or interference from the tower or the RoLi platform, as no particles are generated and emitted during RoLi's operation.



One detail to mention is that, although the profiles shown in Figure 4 and Figure 5 are visualized as if they were measured simultaneously across all heights, the values measured at 8 meters and at 318 meters are always 15 minutes apart. For better visualization, the profiles are plotted as if they were recorded instantaneously. However, most atmospheric parameters, including aerosol data, rarely change over a time span of just a few minutes; therefore, this way of visualization has been chosen.

## 4 Conclusions

This study presents the design, construction, and exemplary data of a newly developed automated lift system for high-resolution vertical profile measurements at the 325 meter tall tower. The platform enables the collection of high-resolution profiles using various aerosol measurement devices in combination with meteorological sensors. By using a single set of instruments, the system minimizes costs associated with deploying multiple instrument sets across the tower and eliminates complications related to instrument comparisons and cross-calibrations. The fully automated and customizable movement patterns of the
system provide flexibility, allowing users to adapt measurements to specific research needs.

The exemplary results demonstrate the system's high reliability and operational robustness, even under challenging rain forest conditions. Equipped with a spacious instrument box and a permanent 230 V power supply, the platform supports a wide range of instruments that can be tailored to user requirements. Additionally, its integration with the local network enables remote access to the instruments and the ability to initiate sampling when needed. Furthermore, the short inlet line of RoLi avoids
the effects and unavoidable losses associated with long inlet lines. For some parameters like wind and temperature, measuring close to the tower structure can produce some measurement artefacts, which have to be considered when analyzing the data. The exemplary results underline the platform's potential for advanced atmospheric measurements. Its high resolution facilitates the detection of thin particle layers and the vertical movement of fog, clouds, and accumulated particles across the rain forest canopy. The data confirm that RoLi is a valuable tool for measurements across a wide range of environmental conditions.
Notably, the system remains operational during rain events and windy conditions, which is crucial for in-depth investigations of atmospheric processes.





## Appendix A: RoLi Software

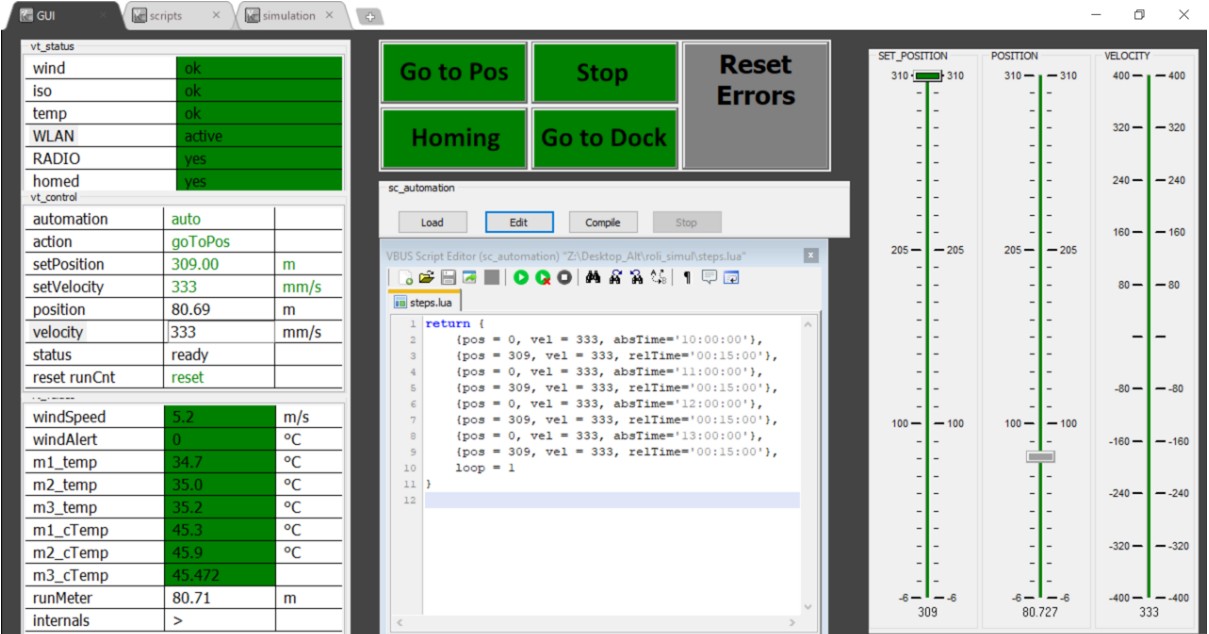

**Figure A1.** Screenshot of the RoLi control software running on the ground-based control computer. The interface features four large square buttons in the upper middle section, used to issue commands to the lift as described in Section 2.6. The top-left panel displays the statuses of critical RoLi parameters, such as wind alarm, rail insulation alarm, motor temperature alarm, WLAN (WiFi) connection alarm, radio connection status, and homing status. If an alarm is triggered or a status changes, the corresponding indicator on the left panel turns red, and the command panels also change color to reflect the issue. The "Reset Errors" button allows operators to reset the alarms after investigating and resolving the cause. The middle-left panel provides a live status overview of RoLi. Automation can be activated here. The "Action" section enables operators to send commands to RoLi, similar to the middle control buttons. The "setPosition" field allows setting the target height, while "setVelocity" specifies the desired vertical speed. The fields "position," "velocity," and "status" display real-time data from RoLi's internal systems, including its current position (relative to the docking station), vertical speed, and overall operational status. The "reset runCnt" button resets the "runMeter" counter located on the bottom-left panel. In the bottom-left section, environmental and operational parameters such as wind speed and motor temperatures are displayed. The current wind speed, as measured by the weather station, is monitored, and a "windAlert" timer starts counting if wind speeds exceed $15\,\mathrm{m\,s^{-1}}$. An alarm is triggered if the threshold is sustained for more than 5 seconds. The six temperature indicators show the temperatures of the three motors and their controllers. The middle section of the interface is used to program RoLi's automation sequence. In this example, RoLi is programmed to go to $0\,\mathrm{m}$ (corresponding to $8.3\,\mathrm{m}$ above ground) and then start measuring at 10:00. It will ascend at a speed of $333\,\mathrm{mm\,s^{-1}}$ until reaching $309\,\mathrm{m}$ ($317.3\,\mathrm{m}$ above ground), wait for 15 minutes, and then descend back to $0\,\mathrm{m}$ at the same speed. RoLi will then remain idle until the next scheduled command at 11:00. On the right side of the interface, indicators display the target position, current position, and velocity of RoLi, providing additional clarity and control to the operator.



*Data availability.* The datasets presented here are available under https://doi.org/10.17617/3.P0PHTC

*Author contributions.* According to Contributor Roles Taxonomy (CRediT, https://credit.niso.org/):

455        Conceptualization: BN, CP

Data curation: SB

Formal analysis: SB

Funding acquisition: UP, PA, CP

Investigation: SB, BN, JDF, DG, FD, CG, TKe, TKl, ML, RV, SW

460        Methodology: SB, CP, BN, TKe, TKl, JDF, CG, ML

Software: CG, ML

Supervision: UP, CP

Validation: BW, UP

Visualization: SB, BN

465        Writing – original draft: SB, CP

Writing – review & editing: All authors

*Competing interests.* The authors declare that they have no conflict of interest.

*Acknowledgements.* We gratefully acknowledge the German Federal Ministry of Education and Research (BMBF, contracts 01LB1001A and 01LK2101B) and the Max Planck Society for their support of this project and the construction and operation of the ATTO site. We also

extend our gratitude to the Brazilian Ministério da Ciência, Tecnologia e Inovação (MCTI/FINEP), as well as the Amazon State University (UEA), FAPEAM, LBA/INPA, and SDS/CEUC/RDS-Uatumã for their contributions to the construction and operation of the ATTO site. We are deeply thankful to all colleagues involved in the technical, logistical, and scientific support of the ATTO project. In particular, we express our heartfelt thanks to Reiner Ditz for his essential contributions to the project's logistics, and to Feliciano de Souza Coelho and Antonio Huxley Melo Nascimento for their assistance with the installation of RoLi. The field experiments and instrumentation group of the MPI for

Biogeochemistry—comprising Olaf Kolle, Martin Hertel, Karl Kübler, and Tarek El-Madany—deserves special acknowledgment for their significant role in the technical integration of the RoLi system into the ATTO infrastructure. I also would like to express my gratitude to Konstantinos Barmpounis for his invaluable assistance with the technical development and testing of RoLi. Furthermore, we thank Sipko Bulthuis, Stefan Wolff, Susan Trumbore, Alberto Quesada, Hermes Braga Xavier, Nagib Alberto de Castro Souza, Thiago de Lima Xavier, Thomas Disper, André Luiz Matos, Antonio Ocimar Manzi, Roberta Pereira de Souza, Wallace Rabelo Costa, Amauri Rodriguês Perreira,

Bruno Takeshi, and Adir Vasconcelos Brandão for their invaluable technical, logistical, and scientific support within the ATTO project. We also extend our gratitude to Jens Weber, Jana Englert, Leslie Kremper, Lena Heins, Cybelli Barbosa, Ricardo H. M. Godoi, Luiz A. T.



Machado and Joseph Byron for their scientific input and stimulating discussions. Bettina Weber acknowledges support from the FWF project CryptXChange (funding no. P 36052-B) and the University of Graz. Sebastian Brill acknowledges the Max Planck Graduate Center with Johannes Gutenberg-Universität Mainz (MPGC). Rafael Valiati acknowledges the support from CAPES and FAPESP.



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
