# Peer review of "Automated atmospheric profiling with the Robotic Lift (RoLi) at the Amazon Tall Tower Observatory"

_EGUsphere, 2025_

## Referee Comment (RC1)

This paper presents the design and construction of a unique system, RoLi, for the tower-based profile studies. Exemplary high-resolution profile data are provided, demonstrating the robust performance and successful application at the ATTO tower with a payload of ~80kg. I can see the future application of this system measuring an expanded number of compounds, maybe even VOCs. The onboard system and rails provide strong support for the scientific instruments in terms of safety control, data acquisition, and time synchronization. While several artefacts were noticed and appropriately addressed by the authors, recommendations are provided to further improve. The capability of this system for the vertical study of fog, clouds, and accumulated particles across the rain forest canopy is clearly demonstrated. Overall, I believe this manuscript suits well with the scope of AMT and is well-written, warranting publication after addressing two concerns and minor technical questions.

1. The aluminum rail in this RoLi system. I don't see a detailed description and evaluation of the strength of this commercial rail system. The Authors stated that "…is ideal for RoLi" but according to Figures 1 and 2, the thickness of the aluminum rail is notably smaller than the supporting frame. Considering the pulling forces from the ~80kg instrument payload, it would be more convincing if the authors could add some evaluation results about the strength. I am sure some words from the manufacturer's specifications would strengthen this claim.

2. Potential contamination of the RoLi system. In this paper, the authors demonstrated the application of RoLi system in the research area related to aerosol profiles. I am sure with such a payload, this system can support measurement of other targets, even VOCs. It would be great if the authors could clarify whether lubricants, solvents, or maintenance materials could introduce interference. Such words would enhance the credibility for multi-purpose future applications.

Below are some more specific comments:

1. Line 105: Insulation is crucial for the tower-based measurements. I am thinking the 100 k$\Omega$ resistance threshold for power supply reactivation described by the authors

might lead to frequent restarts when the unstable resistance fluctuates around 100 kΩ. Maybe a higher restart resistance threshold would be better to prevent unnecessary power cycling and reduce hardware stress.

2.  Line 155: My experience with Raspberry Pi won't allow me to trust such a small processor. The connection between the RoLi system and the ground station is essential since the loss of connection would trigger the alarm. I don't know how frequently the connection check was scheduled (which should be mentioned), but I have double that this Raspberry Pi can handle such a burden of work.

3.  Line 164: I understand that the selection of Cyberbajt YAGI 24-16-2.4 was well discussed by authors and technicians, but some descriptions would be great to show the capability of the effective range and signal strength, etc.

4.  Line 221-226: When there is heavy wind with wind speed exceeding 15 m/s, the contingency plan is to send the RoLi system "go to home", as stated by the authors. My point of view is that staying where it is would be safer. During situations with heavy wind or lost connection, either moving upward or downward is not safe, especially when the connection alarm is triggered, which would send the system home, but if there is something wrong with the rail lower than the current system location, the system is out of control to stop, leading to a dangerous situation. Please correct me if there are unmentioned advantages of the current plan.

5.  Line 242: Overheating inside the box is always painful. My recommendation is to use aluminum-coated foam wrapping around the box. It works great based on my experience.

6.  Line 299: It would be more informative if the authors could provide the list of instruments or the compounds being measured permanently at the top of the ATTO tower.

7.  Line 381-382: I am sure the authors had done that, but are there other parameters measured at the top of the ATTO tower that can help with the analysis?

8.  Line 407-409: A small K-type thermocouple works great in the described situation. Lightweight and adaptable. So thin that it won't be bent by heavy wind. You can set it away from the box and the frame to avoid interference.

---

## Author Comment (AC1)

*Reviewer comments in black, answers in blue and changed text passages in black/red*

**RC1 (18.04.2025)**

Overall, I believe this manuscript suits well with the scope of AMT and is well-written, warranting publication after addressing two concerns and minor technical questions.

*We appreciate the reviewer's positive evaluation of the manuscript and the valuable hints and recommendations provided.*

1. The aluminum rail in this RoLi system. I don't see a detailed description and evaluation of the strength of this commercial rail system. The Authors stated that "...is ideal for RoLi" but according to Figures 1 and 2, the thickness of the aluminum rail is notably smaller than the supporting frame. Considering the pulling forces from the ~80kg instrument payload, it would be more convincing if the authors could add some evaluation results about the strength. I am sure some words from the manufacturer's specifications would strengthen this claim.

   *Thanks for this comment. We have clarified the specifications of the rail in our response here and in the manuscript text. The aluminum rail is a commercial product provided by the company HighStep Systems AG (https://www.highstepsystems.com/en/climbing-system-and-fall-protection/highstep-rail/#technische-daten). According to the manufacturer's specifications, the rail has a stiffness rating of 5 tons per 6-meter segment. This rating ensures that the rail can safely support the dynamic and static loads associated with the RoLi system, including the ~80 kg instrument payload. Although the rail appears thinner than the supporting frame in Figures 1 and 2, it is specifically engineered to provide high strength-to-weight performance. We have included this information now in the revised manuscript to support our claim that the rail is indeed well-suited for the RoLi application.*

   *Line 95 to 98:*

   *... The rail consists of 6 m sections, each secured to the tower structure every 3 m, and is engineered for high strength-to-weight performance. It has a stiffness rating of 5 tons per 6-meter segment, sufficient support both static and dynamic loads of the RoLi system, including the approximately 80 kg instrument payload. This commercial rail system is ideal for RoLi, as it has been easy to install and maintenance-free. ...*

2. Potential contamination of the RoLi system. In this paper, the authors demonstrated the application of RoLi system in the research area related to aerosol profiles. I am sure with such a payload, this system can support measurement of other targets, even VOCs. It would be great if the authors could clarify whether lubricants, solvents, or maintenance materials could introduce interference. Such words would enhance the credibility for multi-purpose future applications.

   *The reviewer points out an important topic: potential contaminations (of any kind) by the RoLi platform itself have been discussed and considered carefully during the development phase. Outgasing compounds, such as lubricants, have been avoided as far*

*as possible. No lubricant is for instance used on the rail itself. For some moving parts, however, lubrication was necessary, which could cause contaminations especially with measurements of volatile organic compounds (VOCs). Specifically, the RoLi system uses high-temperature grease on moving components, which could under certain conditions be a potential source of VOC contamination. For upcoming VOC measurements, targeted tests are planned to identify such potential interferences. Constructive changes on the system, such as a shielding of certain parts or relocation of the inlet, might be required. Alternative lubricants could be considered as well, if the currently used substances turn out to cause interferences.*

*Line 450 to 455:*

*...Furthermore, none of the aerosol and ozone data show any measurement artifacts or interference from the tower or the RoLi platform, as no particles are generated and emitted during RoLi's operation. For upcoming VOC measurements, targeted tests are planned to identify potential interferences from the RoLi platform itself, such as emissions from lubricants or other construction materials. Modifications to the system, such as shielding of specific components or relocation of the inlet, might be required. Alternative lubricants, such as low-emission greases, could also be considered if the currently used substances are found to introduce contamination. ...*

Below are some more specific comments:

1. Line 105: Insulation is crucial for the tower-based measurements. I am thinking the 100 kΩ resistance threshold for power supply reactivation described by the authors might lead to frequent restarts when the unstable resistance fluctuates around 100 kΩ. Maybe a higher restart resistance threshold would be better to prevent unnecessary power cycling and reduce hardware stress.

   *We have indeed considered adjusting the insulation threshold to reduce the potential for frequent shutdowns due to transient fluctuations around the 100 kΩ value. However, regular cleaning of the algae and dirt buildup on the power rails has proven effective in maintaining sufficient insulation and preventing undesired power interruptions. While lowering the insulation threshold might reduce the number of shutdowns, it could compromise safety by weakening the ground fault protection. Since insulation-related power losses occur mainly during heavy rain events and remain rare overall, we have opted to retain the 100 kΩ threshold to ensure operational safety without introducing unnecessary risks.*

2. Line 155: My experience with Raspberry Pi won't allow me to trust such a small processor. The connection between the RoLi system and the ground station is essential since the loss of connection would trigger the alarm. I don't know how frequently the connection check was scheduled (which should be mentioned), but I have double that this Raspberry Pi can handle such a burden of work.

   *The connection monitoring system sends a keepalive signal once per second (info added to manuscript). This interval is configurable via software. During extended field operation, we have not experienced any false positives or missed detections. Given that RoLi's maximum movement speed is 0.3 m/s under normal conditions, any connection loss would be detected within approximately 30 cm of travel. It is also important to clarify that the connection to the ground station is primarily used for transmitting new positioning*

*commands and data transfers. All real-time positioning, movement execution, and monitoring are handled autonomously by the onboard Raspberry Pi in conjunction with the motor controllers. Furthermore, key safety functions, such as emergency stop and overspeed protection, are directly implemented on the motor controllers themselves, providing an additional layer of independent protection in the event of Raspberry Pi failure.*

3. Line 164: I understand that the selection of Cyberbajt YAGI 24-16-2.4 was well discussed by authors and technicians, but some descriptions would be great to show the capability of the effective range and signal strength, etc.

   *The Cyberbajt YAGI 24-16-2.4 is a directional 2.4 GHz antenna with 16 dBi gain, well suited for outdoor point-to-point or point-to-multipoint communication over moderate distances. Supporting standard WiFi protocols (IEEE 802.11b/g/n) and built with weather-resistant materials, it ensures stable performance in harsh environments. Under ideal conditions with clear line of sight (which is the case between the RoLi ground station and RoLi on the tower), it can reach distances of up to ~7 km, making it a reliable option for wireless data transmission in field deployments (info added to manuscript).*

   *Line 172 to 175:*

   *... RoLi itself is equipped with a long-range Wi-Fi antenna (Cyberbajt YAGI 24-16 - 2.4 GHz Wi-Fi Directional Yagi Antenna, Cyberbajt, Młyńska 27, 22-400 Zamość, Poland)* *offering a reliable line-of-sight range of up to 7 km* *and ensuring stable communication and data transmission between the RoLi system and the ground station under outdoor field conditions (Figure 1). ...*

4. Line 221-226: When there is heavy wind with wind speed exceeding 15 m/s, the contingency plan is to send the RoLi system "go to home", as stated by the authors. My point of view is that staying where it is would be safer. During situations with heavy wind or lost connection, either moving upward or downward is not safe, especially when the connection alarm is triggered, which would send the system home, but if there is something wrong with the rail lower than the current system location, the system is out of control to stop, leading to a dangerous situation. Please correct me if there are unmentioned advantages of the current plan.

   *To ensure safe operation, the RoLi system is designed to return to the ground automatically in potentially hazardous conditions. If a wind or connection alarm is triggered, for example during sudden gusts, the system immediately begins a downward movement at maximum speed (0.33 m/s), minimizing the time spent at higher elevations where wind loads are greater. Keeping the system elevated during strong winds could cause the lift to swing, potentially placing mechanical stress on both the rail and the lift itself. The wind alarm is configured to trigger already under conditions that are still considered safe, acting as a preventive measure to avoid exposure to unsafe wind speeds.*

   *In addition to this automatic safeguard, an operator is always present during RoLi operations and continuously monitors weather conditions. If thunderstorms or strong winds are expected, the operator proactively lowers the system in advance.*

   *Even during automatic descent, RoLi's built-in motor safety features remain active and take priority. If the system encounters any obstruction or abnormal mechanical load, it*

*stops immediately to prevent damage. In addition, an emergency power-off button is installed at the tower base next to the power supply. This allows the operator to immediately stop RoLi under any circumstances, providing an additional level of manual control, even when radio and WiFi connection to RoLi is lost. Finally, the rail system is regularly inspected to ensure its integrity, adding a further layer of operational safety. Taken together, we believe the current "go to home" procedure represents a safe and reliable strategy for handling adverse weather conditions. The manuscript text has been revised as follows:*

*Line 222 to 248:*

*To further enhance safety during operation, especially when the lift is in automatic mode, several safety features have been implemented in the control software and on RoLi:*

- ***Wind Alarm:*** *Wind data from the onboard weather station are continuously monitored to detect such situations. If the wind speed exceeds 15 m/s for more than five seconds, a wind warning is automatically triggered in the control software. This action stops any ongoing automatic operation, disables user inputs, and immediately issues the "go to home" command.* *The system then descends at a speed of 0.33 m/s, minimizing the time spent at higher elevations where wind loads are more severe. Keeping the system elevated during strong winds could cause the lift to swing, potentially placing mechanical stress on both the rail and the lift structure. The wind alarm is configured to activate already under conditions that are still considered safe, serving as a preventive measure to ensure RoLi is not operating at height during unsafe wind conditions. In addition to this automatic safeguard, an operator is always present during RoLi operations and continuously monitors the weather. If thunderstorms or strong winds are expected, the operator proactively lowers the system in advance, providing an additional level of safety and control.*
- ***Connection Alarm:*** *If the Wi-Fi connection to the ground station is lost, within one second a connection error is triggered and RoLi automatically drives to the home position at a speed of 0.05 m/s, and all user inputs, as well as automatic mode, are deactivated.*
- ***Temperature Alarm:*** *If the temperature of one of the motors or motor controllers exceeds 70 °C, a temperature warning is triggered. Above 60 °C, the cooling fans mounted on top of the motor control unit are activated to prevent further temperature increases. During normal operation, the temperatures of the motors and motor controllers always remained below 70 °C. If the temperature warning is triggered, the lift immediately stops all operations and locks in place, waiting for the motors to cool down.*

*Even during automatic descent (during normal operation and in case an alarm is triggered), RoLi's built-in motor safety features remain active and take priority. If the system encounters an obstruction, such as dirt accumulation or an object blocking the rail, the onboard electronics detect the resulting increase in motor current and torque, immediately stop the movement, and disable user inputs to prevent damage. An emergency power-off button is also installed at the tower base next to the power supply. This allows the operator to stop RoLi immediately under any circumstances, providing an additional level of manual control, even when radio and Wi-Fi connection to the system is lost. Finally, the rail system is regularly inspected to ensure its integrity, adding a further layer of operational safety.*

5. Line 242: Overheating inside the box is always painful. My recommendation is to use aluminum-coated foam wrapping around the box. It works great based on my experience.

*Many thanks for this suggestion. Aluminum-coated foam wrapping could indeed be an additional measure to prevent overheating. Our ventilation system, combined with a double-roof design, has proven to be sufficiently effective so far, even under hot and sunny conditions. In practice, the internal temperature of the box - with all instruments and pumps running - remained only about 5 °C above ambient temperature.*

6. Line 299: It would be more informative if the authors could provide the list of instruments or the compounds being measured permanently at the top of the ATTO tower.

*We agree that providing more detail on the permanently installed instruments adds clarity and context. At different inlet heights along the ATTO tower, a range of atmospheric parameters has been continuously monitored. These parameters include aerosol number size distributions (measured with an SMPS, ~10–400 nm), aerosol mass concentrations ($PM_2.5$ and $PM_{10}$), black carbon (BC) mass, ozone ($O_3$), volatile organic compounds (VOCs), carbon dioxide ($CO_2$), and basic meteorological variables such as temperature, humidity, wind speed, and direction (specifications added to manuscript). This information is provided in the manuscript in the following sections:*

*Line 41 to 45:*

*... To capture the vertical exchange and processing of gases and aerosols, measurements are typically conducted at multiple inlet heights. At ATTO, greenhouse gases are currently measured at six heights (4, 42, 81, 150, 273, 321 m), volatile organic compounds (VOCs) at four heights (i.e., 40, 80, 150, and 321 m; (Zannoni et al., 2020; Pfannerstill et al., 2021; Ringsdorf et al., 2024), reactive species such as ozone at eleven heights (i.e., 0.05, 0.5, 4, 12, 24, 36, 53, 79, 80, 150 and 320 m), and aerosols at two heights (i.e., 60, and 325 m, (Andreae et al., 2015; Franco et al., 2024; Machado et al., 2024)). ...*

*Line 313 to 317:*

*... The 15-minute waiting time at the top of the tower allowed the motors to cool down and provided an opportunity to compare the RoLi instrument data with measurements from permanently installed instrumentation at the top of the ATTO tower.* *These include continuous measurements of aerosol number size distributions (approximately 10–400 nm, SMPS), aerosol number concentration, aerosol mass ($PM_{2.5}$ and $PM_{10}$), black carbon, ozone, VOCs, $CO_2$, and basic meteorological parameters, enabling effective intercomparison and calibration of the RoLi system**. ...*

7. Line 381-382: I am sure the authors had done that, but are there other parameters measured at the top of the ATTO tower that can help with the analysis?

*We are currently working on an in-depth analysis of all the data measured with RoLi over nearly two years, which will be presented in several follow-up studies. The main aim of this manuscript is to demonstrate the functionality and potential of the RoLi platform in detecting and characterizing transient features in the vertical aerosol profile.*

8. Line 407-409: A small K-type thermocouple works great in the described situation. Lightweight and adaptable. So thin that it won't be bent by heavy wind. You can set it away from the box and the frame to avoid interference.

*A small K-type thermocouple is indeed a good option for this application due to its low weight, flexibility, and minimal wind resistance. Placing it slightly away from the box and supporting structures to avoid heat interference is a helpful idea and will be considered for future improvements of the setup. Many thanks for this suggestion.*

---

## Author Comment (AC2)

*Reviewer comments in black, answers in blue and changed text passages in black/red*

**RC2 (30.04.2025)**

Dear authors, congratulations on an interesting paper about this relevant measurement technology. In general, the article is well presented, and the topic is interesting. Although I certainly appreciate articles focused on the development and engineering of measurement systems, it must be stated that they often precede the scientific articles using data collected by these systems. Therefore, the system description articles are often a little thin on the scientific results, focusing their science mostly on the validation and trustworthiness of the newly developed system. Therefore, my review is based on these premises.

*We thank Reviewer 2 for the helpful and constructive comments. All points were carefully considered, and the manuscript has been revised accordingly. We believe the changes have improved the quality and clarity of the manuscript.*

Specific Comments:

1. Abstract Line 6 (and article lines 224, 226, and 298): In the abstract, you describe the RoLi's profiling speed using the more traditional SI unit of m/s. However, in lines 224, 226, and 298, you use mm/s, including a negative sign in line 224. I recommend making these units follow a standard, ideally m/s.

   *We agree with this suggestion and have updated all references to the RoLi profiling speed to consistently use meters per second (m/s) throughout the manuscript, including the abstract and lines 224, 226, and 298. The negative sign in line 224 has also been removed for clarity as it is already clear from the context that the movement is downward, so the negative sign is not necessary.*

2. Line 26: This sentence has a strange wording. Is the tower "AT" the core of the project?

   *The sentence has been changed to:*

   *Line 25 to 27:*

   *"The central infrastructure of the project is the 325 m tall tower, which enables continuous measurements of atmospheric parameters at the forest–atmosphere interface and throughout the lower planetary boundary layer (PBL)."*

3. Line 34: The wording here may need to be improved because it is confusing. Although you offer a reference, I still fail to follow the logic of the argument. How does a large "footprint" enable the integration of regional processes? Would not a network of small flux towers better represent mesoscale and regional processes?

*We agree that the original sentence could be further clarified. The key point is that, due to their height, tall towers are able to sample air masses that have traveled over larger areas and are more mixed, effectively integrating signals from broader spatial scales. In contrast, smaller flux towers typically capture processes at the ecosystem or local scale due to their lower measurement height and more limited footprint. We have revised the sentence to clarify this distinction, which read now as follows:*

*Line 33 to 35:*

*"Complementary to smaller flux towers, which measure at or above canopy height and have a limited footprint area, tall towers have significantly larger footprints and thus capture atmospheric signals that reflect spatially integrated processes over broader regions."*

4. Line 47: The four-dimensional argument here is overstated. Even though the RoLi is increasing resolution in the vertical dimension, it is still attached to a single tower (i.e., a single point in the region) and not a network of tall towers. This needs to be reworded not to overstate the impact of this system.

   *We agree that the original wording could be misinterpreted. Our intention was not to suggest that the RoLi system provides four-dimensional atmospheric data. Rather, the sentence aimed to highlight the general limitation of traditional tower-based measurements at a few fixed inlet heights, which offer only limited insight into inherently four-dimensional atmospheric processes (i.e., varying in time and three-dimensional space). To avoid any misunderstanding, we have revised the sentence in the manuscript to clarify this point.*

   *Line 45 to 49:*

   *However, observations at only a few fixed inlet heights are subject to experimental limitations, as they provide a simplified view of the complex spatial and temporal dynamics involved in the atmospheric exchange of gases and aerosols. In particular, aerosol measurements are often restricted to just one or a few vertical levels, mainly due to the technical challenges and limited flexibility associated with the installation of stainless steel inlet lines.*

5. Line 57: The argument regarding UAS flight time is incorrect. In your scale of profiles, every 30 minutes with 0.5 m/s ascent speeds, there are several commercially available UAS that can easily achieve those flight performances. I believe the real argument against UAS use here is the pilot/or human oversight requirements and the current limitations on availability of battery recharging/swapping technologies.

   *It is true that certain commercially available UAVs are capable of achieving vertical profiling at the speeds and intervals described. Our intention was not to imply a general technical limitation in terms of ascent speed or flight performance, but rather to highlight practical constraints related to continuous or long-duration deployments, such as battery*

*and payload limitations. In contrast, our robotic lift system enables flexible vertical profiling at various speeds, including intermittent stops for sampling or static measurement, while operating autonomously over extended periods without the need for repeated battery swaps or pilot supervision. We have revised the sentence to more accurately reflect this distinction.*

*Line 56 to 58:*

"*Vertical profile measurements using UAVs eliminate the need for long inlet lines but are constrained by limited payload capacity, relatively short flight durations due to battery limitations, and the requirement for human oversight.* "

6. Obs: The best argument for your system is around line 50 - 55 (the short inlet and the measurement accuracy)!

   *Thanks for underlining this point.*

7. Line 326: Here, you mention the nocturnal Low-level jet as "clearly visible"; Well, it is not. In part because the x-axis only has date and not time, so I can at best only estimate day and night, but also because with data for four days, the stronger winds could be associated with other features, such as the rains described. Therefore, I suggest rewording this explanation and improving the figure to mark sunrise and sunset for each day.

   *Thank you for this comment. The nocturnal low-level jet (LLJ) is a well-documented feature in the central Amazon, and we believe the wind profile data presented clearly reflect this phenomenon, particularly given the absence of downdrafts or convective outflows during these periods. To improve clarity, we revised the x-axis to include both date and time and updated the figure caption of Figure 4 and 5 to explicitly state that the shaded areas in Panel A represent nighttime conditions.*

8. Line 334 - 336: Similar to the comment for line 326, this daily cycle is also not "clearly highlighted".

   *X-axis and description of Fig. 4 and 5 has been modified as stated in the answer to comment 7.*

9. Line 346 - 358: The wording for this paragraph is a bit strange. You start describing the behavior of particles, then transition to relating it to wind shear, only to return to addressing the particles, making it confusing. This is particularly true in line 352 when you start a sentence with "The origin of these particles" after talking about winds. Therefore, it is unclear to the reader which particles are referred to by "these".

*We improved the clarity and logical order in that sentence. The sentence read now as follows:*

*Line 365 to 371:*

*Interestingly, this particle layer was poorly detected by the stationary measurements at 60 m and 325 m, as these heights were either below or above the layer (Figure 5 B). The majority of particles within this elevated layer were in the <30 nm size range. Wind direction data indicate a pronounced vertical wind shear during the same period. The layer above the canopy, up to approximately 100 m, was dominated by northwestern winds, whereas winds above 100 m predominantly originated from the northeast. This wind shear may have contributed to the accumulation or formation of the observed particle layer. However, the origin of the <30 nm particles remains uncertain, as black carbon (BC) data from 60 m were unavailable during this period. The BC data from 325 m do not indicate elevated concentrations, suggesting a predominantly natural origin.*

10. Lines 359 - 367: I particularly liked that you addressed system limitations and included a section for it later. Good Job!

    *Many thanks.*

11. Figures 4 and 5: Improve the x-axis to include time and add markers for sunrise and sunset to make day-night cycles clearer.

    *This issue was already addressed in our responses to comments 7 and 8. As a result, we have modified the x-axis of Figures 4 and 5 to include both date and time, and we have updated the figure caption to clarify that the grey shaded boxes in panel A represent nighttime periods, thereby making the day–night cycles more easily identifiable.*

12. Section 3.4 is very good. Good job!

    *Thank you very much for your positive feedback on Section 3.4.*

13. Line 444: You conclude that "the short inlet line of RoLi avoids the effects and unavoidable losses associated with long inlet lines". Which is a confusing sentence because you are saying it avoids the unavoidable (even though the "unavoidable" adjective is attached to the long lines). Additionally, you make this conclusion without having shown a clear comparison between the short and long line systems. Therefore, I do not believe this is a conclusion that the data shown in the article supports. You have to either add the supporting data or remove the conclusion.

*Many thanks for pointing this out. While we do not present a direct comparison with long inlet systems in this manuscript, it is well established in aerosol science that particle losses due to diffusion increase significantly with inlet length, especially for ultrafine particles. We modified the sentence as follows:*

*Line 472 to 474:*

*"... Furthermore, the short inlet line of RoLi helps to minimize particle losses and sampling artifacts that are commonly associated with longer inlet lines, particularly for ultrafine particles".*

Conclusion: Is this system open-source? Can others use it? If so, be sure to highlight it as it would greatly increase the value of this article to the community.

General Recommendation:

1. The system is interesting, and the article is good. However, I believe it could be improved by including more intercomparison data to validate the performance of the RoLi system. As is, the data presented only indicates a minimal agreement with the expected atmospheric patterns of the region for that time of year, without a validation of the instances in which it does not.

   *Thank you very much for your positive overall assessment and your constructive suggestion. We agree that including more intercomparison data would further strengthen the validation of the RoLi system. While the current manuscript focuses primarily on demonstrating the technical capabilities and profiling potential of the system through selected case studies, we acknowledge that a more comprehensive performance evaluation would be beneficial.*
   *The RoLi system has demonstrated reliable performance under challenging rainforest conditions, including high humidity, strong winds, heat, and heavy rain. While the presented data reflect expected patterns in several cases, this study focuses on the technical capabilities of RoLi for high-resolution vertical profiling. A follow-up study is currently in preparation to explore the atmospheric processes in greater detail.*

2. Although I understand that a 330 m tower does not fit in a controlled chamber, and there are not many other towers around, I believe the authors could have demonstrated the performance of the measurements against the point measurements of the tower, the inferred profile from the tower point measurements, and a NWP simulation.

   *We agree that comparison with reference measurements is important. During the RoLi campaigns, two fixed measurement heights for aerosol data (at 60 m and 325 m) were available on the ATTO tower. These data have been included in the manuscript and are shown in the relevant figures for direct comparison with the vertical profiles recorded by RoLi.*

*While this limits the extent of profile-based validation during the campaign, we believe the inclusion of these reference points still provides meaningful context to assess the plausibility and representativeness of the RoLi measurements. We have clarified this in the manuscript. Additional intercomparisons, including model simulations (e.g., from NWP), are part of our planned future work.*

3. Finally, it is unclear if others in the community can benefit from this development. Can I also build a RoLi, or is it proprietary? Can I access and use RoLi data? If not, why should I care about it? Please make these answers clear to the reader. I believe doing so will elevate the impact of this article in the community.

   *RoLi is an open-source platform, and all essential technical details required to build and operate the system are described in this manuscript. For further guidance or implementation support, interested researchers are welcome to contact us directly. We are happy to share additional documentation and assist with adaptations for specific use cases.*
   *Given the growing number of tall research towers worldwide, and the increasing need for vertically resolved measurements of aerosols, trace gases, and VOCs, we believe that the RoLi concept offers a valuable and scalable solution for atmospheric research.*
   *In addition, all RoLi data will be made publicly available through an open-access repository once the associated manuscripts are published, ensuring transparency and enabling further use by the research community.*

---

## Author Response (AR2)

Dear Dr. Wang, dear editorial team,

many thanks for handling our manuscript entitled "Automated atmospheric profiling with the Robotic Lift (RoLi) at the Amazon Tall Tower Observatory". Many thanks also for your positive evaluation.

We would like to respond here to the "editor decision" comment in the EGUSPHERE manuscript system. You commented there:

"Editor comment (EC): I would like to accept this interesting work, subject to one referee's concern that while the paper has concentrated on describing the concept, it lacks sufficient data, thus requiring a more conservative tone in the abstract and other sections."

Author comment (AC): Many thanks for the positive overall evaluation of our manuscript. We suppose that you refer to the following comment be referee #2:

Referee comment (RC): "Although I certainly appreciate articles focused on the development and engineering of measurement systems, it must be stated that they often precede the scientific articles using data collected by these systems. Therefore, the system description articles are often a little thin on the scientific results, focusing their science mostly on the validation and trustworthiness of the newly developed system."

AC: We agree with your assessment here that technical manuscripts, like this one, (should) lean towards a detailed description of instrumentation, rather than a comprehensive analysis of data sets. In this particularly case, we aimed for the right balance between showing and discussing enough data to validate the capability of the RoLi system, however, to not anticipate results for the follow-up RoLi research papers. In order to address your concern that conclusions might be drawn on a comparatively thin data basis, we revised the relevant statements, aiming for a more "conservative" tone. In the abstract, the following statement has been revised:

> "First measurement results show pronounced spatiotemporal patterns in the altitude profiles of temperature, humidity, fog, and aerosol particle concentration and size, providing new insights into the diel interplay of convectively mixed daytime and stable stratified nighttime conditions."

which reads now as follows:

> "First measurement results show spatiotemporal patterns in the altitude profiles of temperature, humidity, fog, and aerosol particle concentration and size. This proves RoLi's technical capability to resolve the diel interplay of convectively mixed daytime and stable stratified nighttime conditions."

In page 14, line 311, we added the following statement to clarify the role and scope of the initial RoLi data presented in the manuscript:

> "The meteorological and aerosol profile data shown in the following paragraphs represent selected parts of a larger data set. The selected profiles are meant to demonstrate the technical capabilities and limitations of the RoLi system. Only selected data are shown here in order not to anticipate subsequent studies. A detailed analysis of the profile data will be subject of follow-up studies."

We hope that this addresses your concerns. We, the author team, are of course available if further changes or modifications of the text are needed.

With best regards,

Sebastian Brill